# Serum Exosomal MicroRNA, miR-10b-5p, as a Potential Diagnostic Biomarker for Early-Stage Hepatocellular Carcinoma

**DOI:** 10.3390/jcm9010281

**Published:** 2020-01-20

**Authors:** Hyo Jung Cho, Jung Woo Eun, Geum Ok Baek, Chul Won Seo, Hye Ri Ahn, Soon Sun Kim, Sung Won Cho, Jae Youn Cheong

**Affiliations:** 1Department of Gastroenterology, Ajou University School of Medicine, Suwon 16499, Korea; pilgrim8107@hanmail.net (H.J.C.); jetaimebin@gmail.com (J.W.E.); ptok99@hanmail.net (G.O.B.); countpas@naver.com (C.W.S.); rhkwp37@naver.com (H.R.A.); cocorico99@gmail.com (S.S.K.); sung_woncho@hotmail.com (S.W.C.); 2Department of Biomedical Sciences, Ajou University Graduate School of Medicine, Suwon 16499, Korea

**Keywords:** hepatocellular carcinoma, exosome, biomarker, miR-10b-5p

## Abstract

Exosomal microRNAs (exo-miRs) have been promising cancer biomarkers. MiRs in hepatocellular carcinoma (HCC) cell-derived exosomes (HEX) were analyzed to identify reliable serum biomarkers for HCC. To detect overexpressed miRs in HEX, extracted exosomal small RNAs from human HCC cell lines and normal hepatocytes were sequenced and analyzed. Clinical significance of the overexpressed miRs in HEX was evaluated using quantitative real-time PCR (qRT-PCR) on serum samples of a validation cohort consisting of 28 healthy individuals, 60 with chronic liver disease, and 90 with HCC. We found 49 significantly overexpressed miRs in HEX compared to a normal hepatocyte. Among them, miR-10b-5p, miR-18a-5p, miR-215-5p, and miR-940 were overexpressed in HCC tissues and also associated with prognosis of HCC in the analysis of a public omics database. qRT-PCR analysis of the four serum exo-miRs in the validation cohort revealed serum exo-miR-10b-5p as a promising biomarker for early-stage HCC with 0.934 area under the curve (AUC) (sensitivity, 90.7%; specificity, 75.0%; cutoff value, 1.8-fold). Overexpression of serum exo-miR-215-5p was found to be significantly associated with poor disease-free survival in patients with HCC. Serum exo-miR-10b-5p is a potential biomarker for early-stage HCC, while serum exo-miR-215-5p can be used as prognostic biomarker for HCC.

## 1. Introduction

Hepatocellular carcinoma (HCC) is the fifth most common malignancy and third leading cause of cancer-related mortality worldwide [1]. Though HCC is known as intractable cancer with poor prognosis, patients diagnosed at a very early stage (tumor size ≤2 cm) have a five-year survival rate of more than 70% after effective therapeutic treatment [2]. As a result, professional societies such as the Association for the Study of Liver Diseases (AASLD), the European Association for the Study of the Liver, and the Asian Pacific Association for the Study of the Liver recommend HCC surveillance every six months for patients at a high risk of developing HCC [3,4]. Currently, abdominal ultrasound with or without serum alpha-fetoprotein (AFP) is used for HCC surveillance. While abdominal ultrasound is recommended in all guidelines, usage of serum AFP as a tumor biomarker remains controversial due to low sensitivity and suboptimal cost-effectiveness in the detection of early-stage HCC [3]. Thus, reliable serum biomarkers are urgently needed for detecting early-stage HCC to improve the prognosis of patients with HCC.

In recent years, liquid biopsy has emerged as a promising technology in cancer biology. Using liquid biopsy, tumor-derived circulating genetic molecules can be investigated by minimal invasion [5]. Exosomes are one of the most important components in a liquid biopsy. These are extracellular vesicles of about 30–100 nm in size, enclosing genetic materials of the parent (original) cell [6]. The exosome delivers the genetic materials from the parent cell to the recipient cell, and thus, it is considered to be the key player in intercellular communication [7]. Many studies have analyzed the molecular contents of exosomes to find efficient biomarkers and therapeutic targets. From the exosomal cargo, microRNAs (miRs) have gained profound attention. Increasing evidence suggests that the loading of miRs into an exosome is not a random process. Incorporation of an actively selected miR into an exosome depends on the characteristic of the parent cell [8].

The objective of this study was to investigate a reliable serum biomarker for patients with HCC by analyzing the sequences of exosomal microRNAs (exo-miRs) derived from HCC cells. Clinical significance of the overexpressed miRs in an HCC cell-derived exosome (HEX) was evaluated by analyzing public omics data and validated using serum samples of an independent cohort with liver disease.

## 2. Experimental Section

### 2.1. HCC Cell Lines and Culture

Human HCC cell lines, Hep3B and Huh-7, were cultured in Dulbecco’s Modified Eagle Medium, which was supplemented with 10% fetal bovine serum in a humidified atmosphere with 5% CO_2_. Immortalized normal hepatocyte cells, THLE-2, were cultured in LHC-8 medium containing 70 ng/mL phosphoethanolamine, 5 ng/mL epidermal growth factor, 10% FBS, and antibiotics at 37 °C in a humidified atmosphere with 5% CO_2_.

### 2.2. Exosome Isolation from Cell Culture Media And Peripheral Blood

On attaining 90% confluency, cells were washed with phosphate-buffered saline and supplemented with DMEM containing 10% exosome-depleted FBS (System Biosciences, PA, USA) for 72 h. Next, culture medium was collected and centrifuged consecutively at 300× *g* for 10 min at 4 °C, at 2000 × *g* for 10 min at 4 °C, and then at 7500 rpm for 20 min at 4 °C to remove cells, dead cells, and cell debris. Furthermore, the supernatants were ultracentrifuged at 30,000 rpm for 70 min at 4 °C to pellet crude exosomes. Pellets were washed twice with phosphate-buffered saline, resuspended in 100 µL PBS, and stored at −80 °C. Additionally, human peripheral blood was collected from patients, left to coagulate for 20 min at room temperature (25–26 °C), and then centrifuged at 1500× *g* for 20 min. The resulting supernatant (serum) was aliquoted in 1 mL tubes and stored at −80 °C for subsequent exosome isolation. Furthermore, 1 mL serum aliquots were thawed at room temperature and exosomal RNA was isolated from serum using SeraMir Exosome RNA Amplification kit (System Biosciences).

### 2.3. Transmission Electron Microscopy

For imaging analysis, exosomes were stained with 10 nm gold-conjugated anti-CD63 antibody. Samples were fixed with 2% glutaraldehyde and 4% paraformaldehyde for 2 h at room temperature. Exosomes were then visualized using a transmission electron microscope (TEM).

### 2.4. MiR Sequencing

Total RNA was extracted from exosomes, and only small RNAs ranging from 18 to 30 nucleotides were used for library construction. Following PCR amplification, products were sequenced using the Illumina HiSeq 2000 system (Illumina Inc, San Diego, CA, USA).

### 2.5. Ingenuity Pathway Analysis (IPA)

Datasets of differentially expressed exo-miRs were analyzed through the use of IPA (QIAGEN Inc., United States, https://www.qiagenbioinformatics.com/products/ingenuitypathway-analysis). Functional analysis of the data revealed the biological functions and diseases that were significantly associated with the dataset. Canonical pathways that were significantly associated with the dataset were analyzed using the IPA library of canonical pathways.

### 2.6. Analysis of Publicly Available Genomic Data

To evaluate the expression level of miR biomarkers in HCC, genomic data were obtained from The Cancer Genome Atlas Liver Hepatocellular Carcinoma (TCGA-LIHC) project. Level 3 TCGA-LIHC miR expression data were log2-transformed [log2(TPM + 1)] to assess the miRNA expression levels.

### 2.7. MiR Target Prediction and Molecular Pathway Mining

An in silico analysis was performed to predict the targets of the filtered four oncogenic miRs using the miRanda database (http://www.microrna.org/microrna/home.do). MiR sequences were obtained from the miRBase database (http://www.mirbase.org). To investigate exo-miR target signatures that were enriched in the known molecular databases, we downloaded gene sets from MSigDB (http://software.broadinstitute.org/gsea/msigdb) using Broad Institute’s Gene Set Enrichment Analysis software (http://www.broadinstitute.org/gsea). The false discovery rate was considered to be statistically significant if *q* < 0.05. To visualize the link between the four exo-miRs and their targets, miRnet (https://www.mirnet.ca/miRNet/faces/home.xhtml) platform was used.

### 2.8. Quantitative Real-Time PCR (qRT-PCR)

Expression of serum exo-miRs was evaluated using qRT-PCR. cDNA synthesis was performed using a miScript RT II kit (QIAGEN). Furthermore, qRT-PCR was performed using amfiSure qGreen Q-PCR Master Mix (Gendepot, TX, USA) and monitored in real time using the CFX Connect Real-Time PCR Detection System (Bio-Rad Laboratories, CA, USA). MiR-1228-3p was used as an internal control. A relative standard curve method (2^−ΔΔCT^) was used to determine the relative expression. All measurements were confirmed three times. Primer sequences used in the study are illustrated in Appendix A. The design and procedure of the present study was approved by the Institutional Review Board of the Ajou University Hospital, Suwon, South Korea (AJRIB-BMR-KSP-18-397). The informed consent was waived.

### 2.9. Validation Cohort and Clinical Term Definitions

Serum samples and the data used in this study were provided by Biobank of Ajou University Hospital, a member of the Korea Biobank Network. Serum samples were collected from patients who visited Ajou University Hospital, Suwon, South Korea between January 2014 and December 2018. The study groups were categorized as normal healthy individuals, patients with chronic hepatitis B (CHB), patients with liver cirrhosis (LC), and patients with HCC. Normal control was defined as a patient aged from 18 to 50 years old who visited Ajou Health Promotion Center for regular health checkups without any medical history and completely normal results. Patients with CHB were defined as having persistence of serum HBsAg for more than six months [9]. Patients with LC were diagnosed based on morphological assessment of imaging study, liver stiffness in elastography, and several blood tests measuring platelet count, albumin level, and international normalized ratio [10]. HCC was diagnosed in patients according to AASLD guidelines [3,4]. The provided clinical data contained information about age, sex, etiology of liver disease, aspartate aminotransferase level, alanine aminotransferase level, platelet count, serum AFP level, serum albumin level, serum bilirubin level, and international normalized ratio. Additionally, tumor size, tumor number, presence of vascular invasion, and tumor stage according to the modified Union for International Cancer Control (mUICC) staging system were investigated in HCC patients [2]. In this study, early-stage HCC was defined as a single lesion of <2 cm in diameter, corresponding to mUICC stage I. Patients at high risk of developing HCC were defined as patients with CHB or LC. Disease-free survival (DFS) was defined as the time from curative treatment to cancer recurrence, while overall survival (OS) was defined as the time from HCC diagnosis to death by any cause.

### 2.10. Statistical Analysis

All experiments were performed at least three times and all samples were analyzed in triplicate. The experimental methods were performed in a blinded fashion. Data are presented as mean ± standard deviation (SD) or standard error of the mean (SEM). Statistical significance of the difference between experimental groups was assessed by paired or unpaired Welch’s *t*-test, Mann–Whitney U test, and ANOVA test. IBM SPSS software version 22.0 (SPSS Inc., Chicago, IL, USA) and GraphPad Prism version 7.01 software (San Diego, CA, USA) were used for statistical analysis. Statistical significance was established at *p* < 0.05. Chi-square test (two-sided) was used to assess the association between categorical parameters. Survival curves were plotted using the Kaplan–Meier method, and significant difference between the survival curves was determined using the log-rank test. Receiver operating characteristic (ROC) curves were analyzed to evaluate sensitivity, specificity, and respective area under the curve (AUC) value with 95% confidence interval (CI) for each candidate biomarker. A binary cutoff value of each exo-miR expression was determined based on the best Youden’s index on the ROC curves.

## 3. Results

### 3.1. Confirmation of HEX and Identification of Overexpressed HCC-Derived Exo-MiRs

At first, we confirmed the isolated exosomes derived from culture media of each cell line. CD63-immunostained TEM revealed that all isolated samples consisted of CD63-positive spherical vesicles about 30–100 nm in size, which confirmed the efficiency of exosome isolation (Figure 1a). Exo-miRs were sequenced and analyzed as illustrated by the pipeline in Figure 1b. Consequently, 49 exo-miRs were detected as predominantly overexpressed miRs in HEX (>1.5-fold) compared to miR expression in THLE-2-derived exosomes (Figure 1c). Figure 1d shows the heatmap of 49 overexpressed miRs in HEX. The heat map revealed the significant overexpression of 49 miRs in HEX compared to their expression in THLE-2-derived exosomes. Network analyses using IPA were performed to detect the specific function of the overexpressed exo-miRs. We found that overexpressed HCC-derived exo-miRs were significantly associated with hepatic carcinogenesis and metastasis (Figure 1e).

### 3.2. TCGA Data Analysis of HCC-Derived Exo-MiRs

Expression of 49 overexpressed HCC-derived exo-miRs was analyzed in the TCGA database. We extracted sequencing data of 50 nontumorous tissues and 363 HCC tissues from the TCGA database. Comparative analysis revealed that 18 miRs out of the 49 miRs were significantly overexpressed in HCC tissues compared to nontumorous tissues (Figure 2a; Appendix A). Furthermore, survival analysis was performed on the 18 overexpressed miRs to filter a potential biomarker. Data revealed that, specifically, four exo-miRs (miR-10b-5p, miR-18a-5p, miR-215-5p, and miR-940) were significantly associated with poor OS in patients with HCC based on the TCGA dataset (Figure 2c). Figure 2b illustrates the difference in expression of the four selected miRs between nontumorous tissues and HCC tissues. The four exo-miRs were found to be overexpressed in HCC tissues compared to their expression in nontumorous tissues (Welch’s *t*-test, *p* < 0.001). Patients with overexpressed exo-miRs (miR-10b-5p, miR-18a-5p, miR-215-5p, and miR-940) showed significantly poor OS compared to OS of patients with low expression. Consequently, the four miRs (miR-10b-5p, miR-18a-5p, miR-215-5p, and miR-940), which were consistently overexpressed in HEX as well as HCC tissues and showed significant prognostic implications, were selected for independent validation study to assess their clinical significance. Additionally, prediction of target genes corresponding to each exo-miR was performed using the miRanda database (Appendix A). The majority of the miRanda-derived target genes were found to be cell cycle-associated genes (Appendix A). miRnet analysis revealed the visual analytics of the four exo-miRs and their interactions with cell cycle-related target genes (Appendix A).

### 3.3. Diagnostic Significance of the Serum Exo-miRs in Cohort

Diagnostic efficiency of the selected exo-miRs was evaluated in a cohort study that comprised 18 healthy individuals and 19 patients with mUICC stage IV HCC. Substantial expression of each serum exo-miR was evaluated by qRT-PCR. Results revealed that all four serum exo-miRs were significantly overexpressed in patients with HCC compared to corresponding exo-miR expression in the control group (miR-10b-5p: 30.83-fold in HCC group vs. 1.29-fold in control group, *p* < 0.0001; miR-18a-5p: 4.13-fold in HCC group vs. 1.19-fold in control group, *p* = 0.0002; miR-215-5p: 6.69-fold in HCC group vs. 1.28-fold in control group, *p* < 0.0001, miR-940: 1.17-fold in HCC group vs. 1.03-fold in control group, *p* = 0.0004) (Figure 3a). AUC of each serum exo-miR was found to be 0.968, 0.842, 0.936, and 0.827, corresponding to miR-10b-5p, miR-18a-5p, miR-215-5p, and miR-940, respectively (Figure 3b). Two serum exo-miRs, which showed >0.9 AUC (miR-10b-5p and miR-215-5p) were also selected for the validation cohort study.

### 3.4. Significance of Serum Exo-miRs in Diagnosing HCC in the Validation Cohort

To determine the optimal sample size, statistical power analysis was performed using G*Power program 3.1.9.4 (Effect size = 0.15; power = 0.95; α error probability = 0.05), and the minimum sample size for sufficient statistical power was revealed as 107 subjects [11]. The study consisted of 178 patients, which included 28 normal healthy individuals (control), 27 patients with CHB, 33 patients with LC, and 90 patients with HCC. Patients with HCC were categorized according to mUICC guidelines for HCC stage as 32 (36%) in stage I, 14 (16%) in stage II, 26 (29%) in stage III, 11 (12%) in stage IVa, and 7 (8%) in stage IVb. Table 1 shows baseline characteristics of the validation cohort.

Figure 4a shows the serum AFP level and expression of serum exo-miR-10b-5p and exo-miR-215-5p based on the stage of liver disease. Serum AFP level was observed to gradually increase with the progression of liver disease (one-way ANOVA, *f*-value = 2.906, *p* = 0.01). Serum AFP level was found to be significantly upregulated in patients with HCC (4529.32 ng/mL) compared to serum AFP level in normal healthy controls (1.71 ng/mL) (*p* < 0.001). However, we did not find a significant difference between the early HCC group (12.8 ng/mL) and high-risk HCC group, which included patients with CHB (17.55 ng/mL) and LC (49.94 ng/mL). Furthermore, even while comparing patients with LC (49.94 ng/mL) and mUICC stage I HCC (12.8 ng/mL), the serum AFP level of patients with LC was revealed to be higher than that of patients with mUICC stage I HCC. We also found that serum exo-miR-215-5p expression levels gradually increased with liver disease progression (One-way ANOVA, *f*-value = 6.01, *p* < 0.001). Subsequently, serum exo-miR-215-5p was observed to be significantly overexpressed in patients with HCC (8.01-fold change) compared to their expression in normal controls (1.96-fold change). However, consistent with the serum AFP levels observed in this study, there was no significant difference in the expression levels of exo-miR-215-5p between the early HCC group (3.97-fold change) and high-risk HCC group (3.94-fold change for CHB and 2.49-fold change for the LC group). Remarkably, serum exo-miR-10b-5p expression was constant, regardless of liver disease progression in nontumorous conditions. On the other hand, HCC group showed a significant increase in miR-10b-5p levels with a notable rise even in early-stage HCC (*** *p* < 0.001). Serum exo-miR-10b-5p expression levels were found to be differentially expressed when comparing nontumorous conditions and tumorous conditions (1.17-fold change for CHB vs. 106.60-fold change for mUICC stage I HCC (*** *p* < 0.001), 1.35-fold change for LC vs 106.60-fold change for mUICC I HCC (*** *p* < 0.001)) (one-way ANOVA, *f*-value = 7.52, *p* < 0.001).

Figure 4b shows AUCs of each serum marker on HCC diagnosis in all patients. AUC of serum exo-miR-10b-5p was 0.932 with 91.1% sensitivity and 75% specificity when the cutoff value was set at 1.8-fold change. Serum exo-miR-10b-5p AUC was observed to be significantly higher compared to that of serum AFP, which showed 0.707 AUC (cutoff value = 20 ng/mL, sensitivity = 37.8%, and specificity = 71.6%), and serum exo-miR-215-5p, which showed 0.723 AUC (cutoff value = 5.25-fold change, sensitivity = 43.3%, and specificity = 88.6%, *p* < 0.001) (Table 2). Serum exo-miR-10b-5p showed 0.946 AUC when distinguishing patients with mUICC stage I/II HCC from the nontumorous group (Figure 4c) and 0.934 AUC (sensitivity = 90.7% and specificity = 75%) on distinguishing patients with stage I HCC from the nontumorous group (Figure 4d). Serum exo-miR-10b-5p revealed a significant diagnostic accuracy for early-stage HCC, while AUC of serum AFP and serum exo-miR-215-5p was 0.542 (sensitivity = 9.4%, specificity = 71.6%) and 0.666 (sensitivity = 25.0%, specificity = 88.6%), respectively (Figure 4d, Table 2).

Subgroup analyses were performed according to gender, which showed imbalances between HCC and control groups in baseline characteristics. Appendix A demonstrate the results of subgroup analyses for males and females, respectively. As a result, serum exo-miR-10b-5p was identified as a more efficient biomarker than serum AFP, regardless of gender.

Additional analysis was performed to assess whether the combination of serum exo-miR-10b-5p and serum AFP could increase the diagnostic efficiency in detecting early HCC (Figure 5a). However, we did not find any significant difference in the diagnostic efficiencies between using only serum exo-miR-10b-5p and using a combination of serum AFP and serum exo-miR-10b-5p. Such findings might be due to the remarkable diagnostic accuracy of exo-miR-10b-5p compared to that of serum AFP.

### 3.5. Diagnostic Efficiency of Serum Exo-miR-10b-5p in Patients at High Risk of Developing HCC

Subgroup analyses were performed to evaluate the efficiency of serum exo-miR-10b-5p to distinguish early-stage HCC from patients at high risk of developing HCC (Figure 5b). AUC of serum exo-miR-10b-5p in distinguishing HCC (all stages) from the high-risk HCC group was found to be 0.925. Furthermore, AUC in distinguishing stage I HCC and stage II HCC was 0.941, while AUC in distinguishing mUICC stage I HCC from patients at high risk of developing HCC was observed to be 0.935 (sensitivity = 90.6% and specificity = 78.3%) at a cutoff value of 1.8-fold change (Appendix A).

### 3.6. Prognostic Implication of the Serum Exosomal miR-10b-5p and Exosomal miRNA-215-5p

Furthermore, we evaluated the prognostic value of serum exo-miRs. Kaplan–Meier analysis revealed that patients with overexpressed serum exo-miR-215-5p had significantly poor DFS compared to patients with low expression of serum exo-miR-215-5p (*p* = 0.02) (Figure 6a). In addition, serum exo-miR-215-5p expression was found to gradually increase with advancement in tumor stage (Figure 6b). Serum exo-miR-215-5p was significantly overexpressed in patients with vascular invasion compared to patients without vascular invasion (*p* = 0.05) (Figure 6c).

Substantial expression of serum exo-miR-10b-5p was observed to be not significantly associated with survival, tumor stage, or presence of vascular invasion.

### 3.7. Comparison between Diagnostic Efficiencies of Serum Exo-miRNA-10b-5p and Serum miRNA-10b-5p

From the validation cohort, eight normal healthy individuals and 10 patients with HCC were randomly selected for comparing the diagnostic efficiency of serum exo-miR-10b-5p and serum miR-10b-5p. Results revealed that serum miR-10b-5p expression was statistically insignificant between the groups. Conversely, serum exo-miR-10b-5p was significantly overexpressed in the HCC group compared to its expression in the nontumorous group (*p* < 0.001). Moreover, AUC of serum exo-miR-10b-5p was found to be 0.975, while AUC of serum miR-10b-5p was found to be 0.675 (Appendix A).

## 4. Discussion

Liquid biopsy is a noninvasive method for detecting cancer biomarkers and determining therapeutic targets by analyzing circulating tumor molecules in body fluid, specifically blood [2]. Considering the risk and/or cost for solid biopsies, liquid biopsy has a remarkable advantage over tissue biopsy. Especially for patients with HCC, development of a reliable biomarker using liquid biopsy is urgently needed as HCC is conventionally diagnosed by typical radiological findings [4]. In this study, our objective was to detect a novel serum biomarker for HCC using liquid biopsy. We detected overexpressed miR profiles in HEX and demonstrated their significance as potential diagnostic and prognostic biomarkers for patients with HCC. Serum exo-miR-10b-5p showed significant diagnostic efficiency in detecting early-stage HCC, while serum exo-miR-215-5p was revealed as a promising biomarker for predicting prognosis in patients with HCC.

To improve the prognosis of patients with HCC, a reliable serum biomarker for diagnosing early-stage HCC is crucial. Currently, detection of early-stage HCC using a serum marker in patients at high risk of developing HCC is challenging. As the majority of patients with HCC are suffering from chronic hepatitis and/or subsequent liver fibrosis, it is difficult to derive a tumor-specific serum marker from a hepatic inflammatory or fibrosis marker. Many previously proposed serum biomarkers for HCC including AFP are elevated, not only in HCC, but also in patients with chronic hepatitis or cirrhosis. Similarly, the present study confirmed that elevation of serum AFP is based on liver disease progression and was unable to differentiate early-stage HCC from CHB or LC. Serum exo-miR-215-5p showed improved efficiency over serum AFP; however, it failed to distinguish between early-stage HCC and chronic hepatitis or LC. In contrast, serum exo-miR-10b-5p was detected as an HCC-specific serum tumor biomarker. Exo-miR-10b-5p expression was not found to be enhanced in CHB or LC; however, a significant increase was observed in patients with HCC, specifically in very early-stage HCC, having a tumor size less than 2 cm in diameter. Thus, exo-miR-10b-5p was able to differentiate early-stage HCC from patients at high risk of developing HCC with significant diagnostic efficiency. These results suggested that the majority of serum exo-miR-10b-5p may be derived from HCC cells, and thus, its expression might be minimally affected by hepatic inflammation and fibrosis.

Many previously reported studies on HCC serum biomarkers have proposed a combination of HCC-derived novel biomarker and serum AFP for improving diagnostic efficiency. Recently, AFP-L3 and des-γ-carboxy prothrombin (DCP) have been suggested as HCC biomarkers [12,13]. However, the reported AUC for AFP-L3 and DCP was found to be about 0.73 and 0.71, respectively. Furthermore, the combination of AFP and AFP-L3 has been shown to enhance AUC (0.83); however, DCP did not further improve the efficiency. In the present study, the combination of serum exo-miR-10b-5p and AFP slightly improved the specificity; however, AFP did not significantly enhance sensitivity and AUC in detecting HCC, compared to that observed with serum exo-miR-10b-5p alone (Appendix A). This could be due to the remarkable diagnostic efficiency of serum exo-miR-10b-5p. Thus, our study revealed that serum exo-miR-10b-5p alone or in combination with AFP can be a potential biomarker for early-stage HCC. However, further validation in a larger cohort would be required for its effectiveness in clinical practice.

MiR-10b-5p is an oncogenic miR in many types of cancers. Several studies have reported that miR-10b-5p promotes cell invasion and metastasis in HCC by suppressing tumor suppressor genes such as cell adhesion molecule 1 (CADM1), matrix metalloproteinases (MMPs), and CUB and Sushi multiple domains-1 (CSMD1) [14,15,16]. However, there have been only a few studies that have proved the clinical significance of circulating miR-10b-5p in patients with HCC. Jian et al. reported that AUC of serum miR-10b was 0.73 for detecting HCC in patients with chronic hepatitis [17]. In the present study, we showed that AUC of serum exo-miR-10b-5p was much higher than that of circulating serum miR-10b-5p. Our results indicate that serum exo-miR-10b-5p is a more significantly efficient diagnostic biomarker than serum miR-10b-5p (Appendix A). Similarly, Wang et al. reported that serum exo-miR-21 was a better diagnostic biomarker for HCC than whole serum-derived miR-21 [18]. These findings show that tumor-derived miRs are enriched with serum exosomes, rather than the whole serum. Although there have been no studies on why a few miRs are overexpressed or found only in exosomes and not in free circulation, increasing evidence has indicated that loading specific miRs into an exosome depends on a selective loading mechanism according to characteristics of the parent cell [8]. These findings suggest that the exosome plays an important role in transporting miRs between cells in the tumor microenvironment. Further study regarding the functional significance of exo-miR-10b-5p compared to free miR-10b-5p in HCC would be required.

MiR-215-5p is reported to act as a tumor suppressor in many types of cancers [19,20,21]. It has been shown to be overexpressed in tissues as well as serum of patients with HCC [22,23,24]. Similarly, this study confirmed the overexpression of miR-215-5p in HCC tissues and elucidated its association with poor prognosis in patients with HCC using the TCGA database. Furthermore, while evaluating the clinical significance of serum exo-miR-215-5p, our study revealed that overexpressed serum exo-miR-215-5p was significantly associated with poor DFS, presence of vascular invasion, and advanced tumor stage. Further studies in a larger cohort are needed to validate the prognostic implication of serum exo-miR-215-5p.

Recently, many studies have been performed on serum exo-miRs to detect novel biomarkers for HCC [25,26,27]. However, the diagnostic efficiencies of the previously evaluated serum exo-miRs have not been satisfactory for clinical application. Compared to the earlier studies, our study showed significant diagnostic efficiency. The strength of our study was the stepwise selection process of the candidate biomarkers. Though the candidate markers were mainly selected after reviewing previous studies, we sorted the candidate biomarkers by analyzing the sequencing data of HEX-derived miRs. Furthermore, they were filtered by integrative analysis using the TCGA database. Suitable screening of biomarkers using notable tumor cell-derived genomic data and their validation using a public omics database are essential for successful development of an efficient liquid biopsy-based biomarker.

## 5. Conclusions

In conclusion, our present study revealed potential biomarkers for HCC using liquid biopsy. Serum exo-miR-10b-5p is a promising biomarker for detecting early-stage HCC, while serum exo-miR-215-5p can be used as a prognostic biomarker in patients with HCC. However, further validation using a larger cohort is needed to confirm the results of our study.

## Figures and Tables

**Figure 1 jcm-09-00281-f001:**
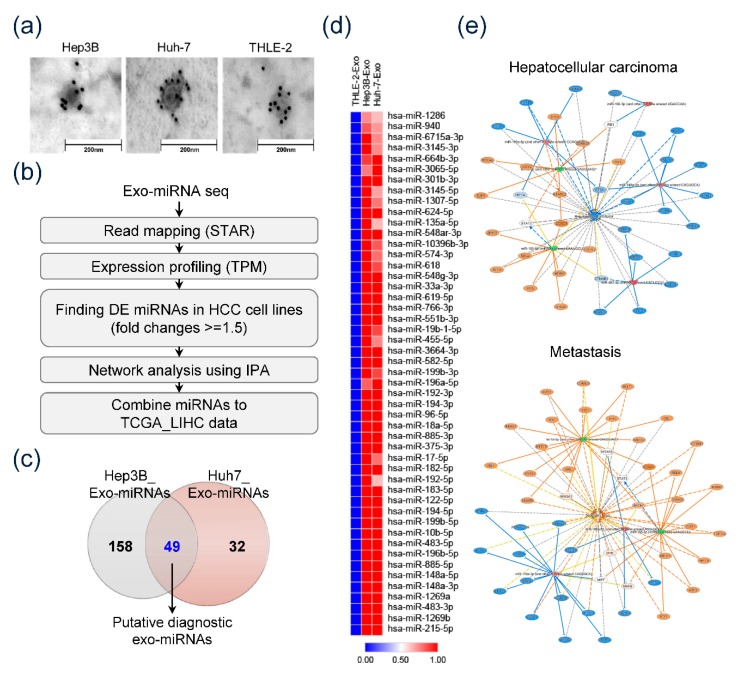
Exosomal microRNAs (exo-miR). Sequencing and detecting specifically overexpressed exo-miRs in hepatocellular carcinoma (HCC). (**a**) Transmission electron microscopy of the exosomes with 10 nm gold-conjugated anti-CD63 antibody. (**b**) Pipeline showing analysis of exo-miRs. (**c**) Venn diagram showing predominantly overexpressed exo-miRs in HCC cell lines compared to expression in THLE-2 cells. (**d**) Heat map depicting 49 predominantly overexpressed exo-miRs in HCC cell lines. Forty-nine miRs seen to be significantly overexpressed in exosomes from Hep3B as well as Huh-7 cell lines; however, not in those of THLE-2 cells. (**e**) Ingenuity pathway analysis of 49 overexpressed exo-miRs of HCC cell lines.

**Figure 2 jcm-09-00281-f002:**
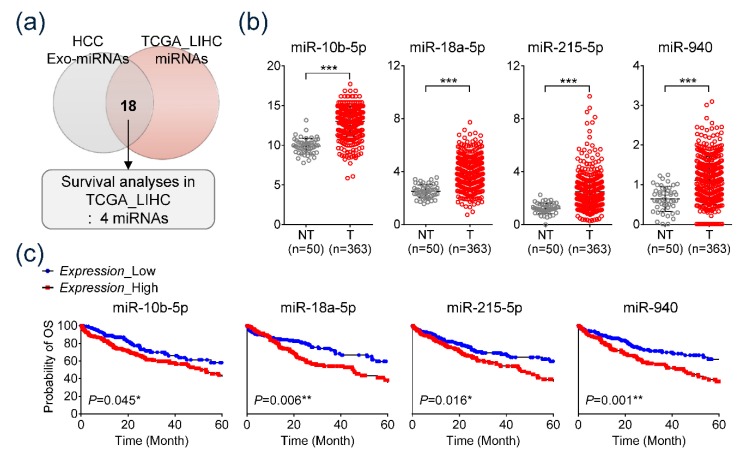
The Cancer Genomic Atlas (TCGA) database analysis, sequencing data of 363 HCC and 50 nontumorous tissues, and corresponding clinical data. (**a**) Venn diagram showing predominantly overexpressed miRs in HCC cell-derived exosomes and HCC tissues of TCGA dataset. (**b**) Comparative analysis of substantial expression of the four miRs including miR-10b-5p, miR-18a-5p, miR-215-5p, and miR-940 between nontumorous tissues and HCC tissues. (**c**) Kaplan–Meier analysis of overall survival based on tissue expressing the four miRs in TCGA database.

**Figure 3 jcm-09-00281-f003:**
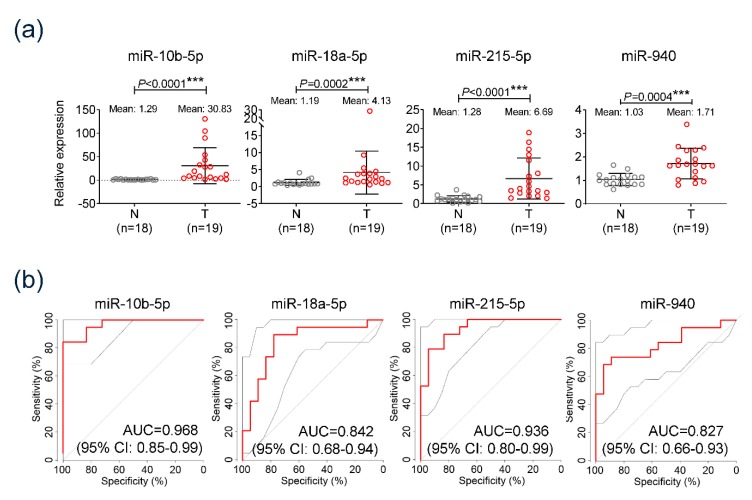
Diagnostic efficiency of serum exo-miRs (miR-10b-5p, miR-18a-5p, miR-215-5p, and miR-940) in diagnosing HCC in test cohort comprising 18 healthy individuals and 19 patients with mUICC stage IV HCC. (**a**) Comparison of expression of the four serum exo-miRs between normal healthy individuals and patients with HCC. (**b**) Area under the curve (AUC) and receiver operating characteristics (ROC) of the four serum exo-miRs in diagnosing HCC.

**Figure 4 jcm-09-00281-f004:**
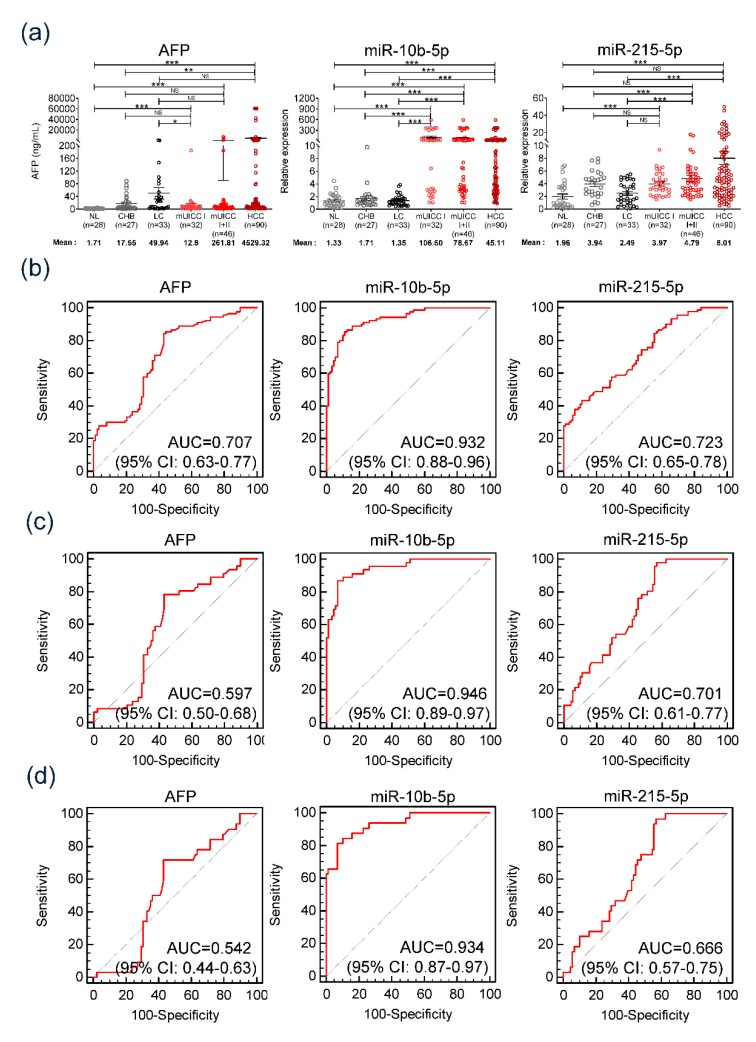
Diagnostic significance of serum alpha-fetoprotein (AFP), serum exo-miR-10b-5p, and miR-215-5p in diagnosing HCC in the validation cohort comprising 28 normal individuals, 27 patients with chronic hepatitis B, 33 patients with liver cirrhosis, and 90 patients with HCC. (**a**) Comparison of serum AFP level and expression of two serum exo-miRs based on stage of liver disease. (**b**) AUC of serum AFP and two serum exo-miRs in diagnosing HCC. (**c**) AUC of serum AFP and two serum exo-miRs in diagnosing mUICC stage I or II HCC. (**d**) AUC of serum AFP and two serum exo-miRs in diagnosing mUICC stage I HCC.

**Figure 5 jcm-09-00281-f005:**
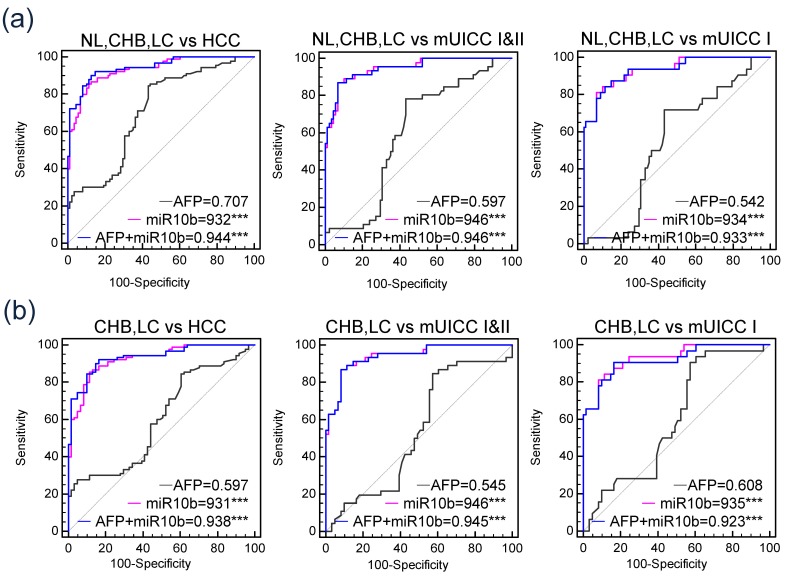
Comparison of diagnostic efficiency of serum AFP, serum exo-miR-10b-5p, and the combination of the serum exo-miR-10b-5p and serum AFP. (**a**) AUC of the serum markers in diagnosing HCC. From right to left: Diagnosing all stages of HCC, mUICC stage I or II, and mUICC stage I HCC, respectively. (**b**) AUC of the serum markers in diagnosing HCC in the subgroup composed of CHB, LC, and HCC. From right to left: Diagnosing all stages of HCC, mUICC stage I or II, and mUICC stage I HCC, respectively.

**Figure 6 jcm-09-00281-f006:**
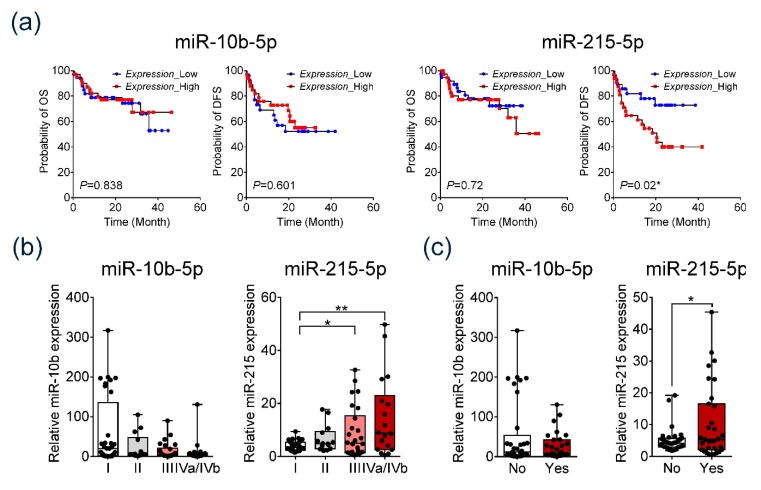
Prognostic significance of serum exo-miR-10b-5p and exo-miR-215-5p in patients with HCC. (**a**) Disease-free survival (DFS) and overall survival analysis based on the expression of the two serum exo-miRs in the validation cohort. (**b**) Expression of the two serum exo-miRs according to the modified Union for International Cancer Control (mUICC) stage guidelines. (**c**) Expression of the two serum exo-miRs based on vascular invasion.

**Table 1 jcm-09-00281-t001:** Baseline characteristics of patients selected for the validation cohort (N = 178).

Variables	Validation Cohort			
	Normal (*n* = 28)	CHB (*n* = 27)	LC (*n* = 33)	HCC (*n* = 90)
Age, mean ± SD	34.1 ± 7.6	45.2 ± 11.1	53.1 ± 10.1	55.2 ± 9.02
Male sex, *n* (%)	2 (7.1)	15 (55.6)	19 (57.6)	70 (77.8)
AST, IU/mL	19.60 ±5.46	50.44 ± 51.50	79.19 ± 100.91	71.57 ± 96.77
ALT, IU/mL	20.00 ± 15.12	57.26 ± 66.31	77.25 ± 100.99	48.12 ± 59.28
Platelet, x109/L	290.40 ± 42.18	190.56 ± 4.39	123.35 ± 65.59	166.44 ± 84.17
AFP (ng/mL), mean ± SD	1.71 ± 0.77	17.55 ± 24.97	49.94 ± 104.09	4290.43 ± 14525.79
Etiology, *n* (%) HBV/HCV/alcohol/others		26 (96.3)/1 (3.7)/0/0	28 (84.8)/3 (9.1)/2 (6.1)/0	82 (91.1)/4 (4.5))/3 (3.3)/1 (1.1)
Albumin (g/L), mean ± SD		4.56 ± 0.42	4.05 ± 0.53	4.28 ± 0.55
Bilirubin (mg/dL), mean ± SD		0.81 ± 0.33	1.05 ± 1.03	1.34± 3.53
INR, mean ± SD		1.23 ± 0.31	1.24 ± 0.11	1.25 ± 0.55
Modified UICC stage I/II/III/IVa/Ivb, *n* (%)				32 (36)/14 (16)/26 (29)/11 (12)/ 7 (8)

CHB, chronic hepatitis B; LC, liver cirrhosis; HCC, hepatocellular carcinoma; ASL, aspartate transaminase; ALT, alanine transaminase; AFT, alpha-fetoprotein; HBV, hepatitis B virus; HCV, hepatitis C virus; INR, international normalized ratio; UICC, Union for International Cancer Control.

**Table 2 jcm-09-00281-t002:** Comparative analysis between using two serum exosomal miR markers and using serum AFP in diagnosing HCC.

**HCC vs. Non Tumor**					
	*P* vs. AFP	AUC	95% CI	Sensitivity (%)	Specificity (%)
AFP (20 ng/mL)	1	0.707	0.635–0.773	37.778	71.591
miR-10b-5p (1.8-fold)	<0.0001	0.932	0.884–0.964	91.111	75.000
miR-215b-5p (5.25-fold)	0.761	0.723	0.651–0.787	43.333	88.636
**mUICC I&II vs. Non Tumor**					
	*P* vs. AFP	AUC	95% CI	Sensitivity (%)	Specificity (%)
AFP (20 ng/mL)	1	0.597	0.509–0.681	15.217	71.591
miR-10b-5p (1.8-fold)	<0.0001	0.946	0.894–0.978	93.478	75.000
miR-215b-5p (5.25-fold)	0.1037	0.701	0.616–0.777	30.435	88.636
**mUICC I vs. Non Tumor**					
	*P* vs. AFP	AUC	95% CI	Sensitivity (%)	Specificity (%)
AFP (20 ng/mL)	1	0.542	0.449–0.634	9.375	71.591
miR-10b-5p (1.8-fold)	<0.0001	0.934	0.874–0.971	90.652	75.000
miR-215b-5p (5.25-fold)	0.1016	0.666	0.574–0.749	25.000	88.636

AFT, alpha-fetoprotein; HCC, hepatocellular carcinoma; AUC, area under the receiving operating characteristic curve; CI, confidence interval; PPV, positive predictive value; NPV, negative predictive value; miR, microRNA; mUICC, modified Union for International Cancer Control.

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
