# Peer review of "Serum Exosomal MicroRNA, miR-10b-5p, as a Potential Diagnostic Biomarker for Early-Stage Hepatocellular Carcinoma"

_jcm, 2020, doi:10.3390/jcm9010281_

Round 1
Reviewer 1 Report
In the article entitled “Serum exosomal microRNA-10b-5p as a potential diagnostic biomarker for early stage hepatocellular carcinoma” by Cho et al., the author has made efforts to identify exosomal miRNAs as a biomarker for the detection of HCC. Serum samples of validation cohort consisted with 28 healthy individuals, 60 with chronic liver disease, and 90 with HCC. The authors found 49 significantly overexpressed miRNAs in HCC cell-derived exosomes (HEX) than that in normal hepatocyte. Among them, miR-10b-5p, miR-18a-5p, miR-215-5p, and miR-940 were overexpressed in HCC tissues, and also associated with prognosis of HCC in the analysis of public omic database. qRT-PCR analysis of the 4 serum exo-miRNAs in the validation cohort revealed serum exo-miR-10b-5p as a promising biomarker for early stage HCC with 0.934 area under the curve (AUC) (sensitivity, 90.7%; specificity, 75.0%; cut-off value, 1.8-fold). Overexpression of serum exo-miR-215-5p was found to be significantly associated with poor disease-free survival in patients with HCC. The authors conclude that serum exo-miR-10b-5p is a potential biomarker for early stage HCC, while serum exo-miR-215-5p can be used as prognostic biomarker for HCC.
This is a good study in the ever-expanding field of miRNAs as biomarkers in cancers.
I have few concerns:
Data presentation seems to be a bit confusing to the general audience. I would suggest to provide an average fold change in addition to the data provided especially in the tables and writeup. Also, the authors should clearly demarcate the differences between exosomal miRNA and circulating serum miRNAs. The authors discuss as to why a few miRNAs were overexpressed or found only in exosomes and not in free circulation. This is interesting as to miR-10b-5p has reversing trend when compared to their presence in exosomes vs serum circulation. It will be interesting to see if miR-10b-5p is upregulated in exosomes or if they have functional significance in HCC (biological significance including tumor promotion).
Overall, this is a good article that will interest audience in HCC and miRNA filed of research. The manuscript will have a considerable impact in the field of cancer.
Author Response
Thank you for your careful review and reasonable comments on our paper. We checked reviewers’ comments and revised one by one.
Please see the attachment

Reviewer 2 Report
Cho and colleauges have performed a study to discover novel exosomal microRNA markers for hepatocellular carcinoma surveillance. The study's strengths include primary discovery, validation in an external public database (TCGA) and validation in a reasonably large cohort of case, high risk controls and normal controls.
The reporting of methods and results could be substantially improved by the following:
-It needs to be stated whether or not the experimental methods were performed in blinded fashion.
-There is no a priori power calculation; this should be provided, especially for the validation study
-Was any correction made for multiple statistical comparisons?
-Section 3.3 needs to provide 95% CI values for the AUCs. For instance, 3/4 plots on figure 3A appear to include the null AUC of 0.5.
-Section 3.4 needs to make adjustments in marker AUCs on the basis of Sex, which appears imbalanced between HCC and control groups.
-More clarity should be provided on the cut-off value (1.8 fold change) used to set sensitivity and specificity for exo-miR-10b-5p.
-PPV and NPV should be deleted from tables 2 and S3. This is not a screening cohort where those values would be best measured. PPV/NPV are dependent on prevalence (of HCC) which is artificially set by the use of a case-control design.
Author Response

(The authors gave the same response as above.)

Round 2
Reviewer 2 Report
The authors have promptly and completely responded to all concerns raised in the prior review.